# Effect of the Mixing Technique of Graphene Nanoplatelets and Graphene Nanofibers on Fracture Toughness of Epoxy Based Nanocomposites and Composites

**DOI:** 10.3390/polym14235105

**Published:** 2022-11-24

**Authors:** Aldobenedetto Zotti, Simona Zuppolini, Anna Borriello, Valeria Vinti, Luigi Trinchillo, Domenico Borrelli, Antonio Caraviello, Mauro Zarrelli

**Affiliations:** 1Institute for Polymers, Composites and Biomaterials, National Research Council of Italy, P.le Fermi, 1, 80055 Portici, NA, Italy; 2Avio S.p.A., Via Leonida Bissolati, 76, 00187 Roma, RM, Italy; 3Sophia High Tech srl, Via Romani, 228, 80048 Sant’Anastasia, NA, Italy

**Keywords:** graphene nanoplatelet (GNP), graphite nanofibers, fracture toughness, nanocomposites, carbon fiber reinforced composite (CFRC), mode I and mode II interlaminar fracture toughness

## Abstract

In this work, the effect of different mixing techniques on thermal and mechanical properties of graphene nanoplatelets (GNPs) and graphene nanofibers (GANFs) loaded epoxy nanocomposites was investigated. Three dispersion methods were employed: a high shear rate (HSR), ultrasonication (US) and the fluidized bed method (FBM). The optical microscopy has revealed that the most suitable dispersion, in terms of homogeneity and cluster size, is achieved by implementing the US and FBM techniques, leading to nanocomposites with the largest increase of glass transition temperature, as supported by the DMA analysis data. The fracture toughness results show a general increase of both the critical stress intensity factor (K_IC_) and the critical strain energy release rate (G_IC_), likely due to the homogeneity and the low scale dispersion of the carbonaceous nanostructures. Based on the nanocomposite fracture toughness improvements and also assuming a potential large scale up production of the nanocomposite matrix, a single mixing technique, namely the FBM, was employed to manufacture the carbon fiber reinforced composite (CFRC). This method has resulted in being less time-consuming and is potentially most suitable for the high volume industrial production. The CFRCs were characterized in terms of tensile, flexural and interlaminar fracture toughness properties and the results were analyzed and discussed.

## 1. Introduction

Carbon fiber reinforced composites (CFRCs) are extensively used in the of aeronautic and aerospace fields, thanks to their outstanding mechanical properties, such as strength and modulus [1]. Nevertheless, the use of CFRCs is still limited due to their low fracture toughness, especially in terms of the interlaminar fracture properties. That weakness is attributed to the highly crosslinked epoxy resins employed as the matrix, which are characterized by a low fracture resistance [2].

A widely used approach for improving the fracture toughness of the epoxy resins consists in the addition of fillers to the system: the increase of the polymer fracture toughness is attributed to the introduction of new mechanisms for the energy absorption during the fracture propagation [3]. In the last decades, different filler typologies were employed to improve the fracture performances of the epoxy matrix, such as inorganic nanoparticles [4], nanoclays [5], rubber particles [6], hybrid nanoparticles [7,8], and carbon nanomaterials [9]. Among these fillers, the most promising candidates to improve the fracture toughness of the hosting matrix are the graphene-based systems, such as graphene nanoplatelets (GNPs), graphene oxide (GO) and carbon nanofibers (CNF).

Worldwide, graphene is one of the most studied materials, thanks to its exceptional mechanical, thermal, and electrical properties. Recently, an increasing number of techniques that allow to produce large quantities of graphene have been developed, thus making cost effective the production of graphene/polymer composites [10]. That class of nanocomposites shows outstanding mechanical properties due to the huge graphene specific surface area, the strong filler/matrix interface and the strength of the sp^2^ carbon bonding network [11].

CNFs are hollow core nanofibers consisting of a single graphite layer or double graphite layers that are stacked parallel or at a certain angle from the fiber axis; they are characterized by a relatively high aspect ratio (>100) [12], with a diameter of about 50–500 nm and a length of a few tens of microns. CNFs have different structures, including bamboo-like, parallel and cup-stacked [13]. They possess high mechanical and excellent physical properties: therefore, CNFs are employed as a cheapest alternative to carbon nanotubes to improve the mechanical, thermal, and electrical properties of the polymer matrix [14].

It is well known that, due to the high surface area and strong van der Waals forces, carbon nanomaterials tend to form large agglomerates. For this reason, the dispersion of these materials in a viscous medium, such as polymer solutions, is a complex challenge. The poor dispersion of the carbon nanomaterials not only significantly lower their efficiency as reinforcement, but also can induce a reduction of the properties, compared to the neat matrix, due to the reduction of the crosslinking density and to the slipping of the nanoparticles when forces are applied to the composites [15]. So the efficiency of the reinforcement is strictly linked to the filler dispersion in the hosting matrix, making the dispersion technique a fundamental parameter in the nanocomposite design [16]. Considerable efforts on the development of the dispersion techniques have been carried out for achieving a homogeneous and well-dispersed system. The most common method used for the manufacturing of epoxy/graphene nanocomposites is the solution mixing [17]: Kang et al. [18] have obtained an increase of the epoxy matrix fracture toughness (93%) by the addition of graphene oxide (1 wt%) previously sonicated in acetone. Furthermore, Rana et al. [19] have employed acetone as a dispersing medium for 0.1 wt% CNF, and they have obtained an increase of the tensile strength and modulus of about 34% and 29%, respectively. However, some solvent-free methods have also been adopted in order to maximize the reinforcing capability of the filler, and consequently the mechanical performances of the epoxy matrix. For example, Wei et al. [20] have realized a graphene based nanocomposite by using a solvent-free method consisting of a hand mixing process followed by a strong sonication phase; this system shows an increase in the tensile strength and the modulus of about 12% and 30%, respectively. Singer et al. [21] have employed the ball-milling process in order to reduce the cluster dimensions and obtained a homogeneous dispersion of CNFs in the epoxy matrix: the flexural modulus and strength result were higher, compared to the neat matrix (14 and 13%, respectively). Munoz et al. [22] have developed a graphene oxide dispersion technique that involves many steps of mechanical dispersion in water and centrifugation, finally followed by a freeze-drying step of the GO suspension: the so obtained aerogel is mechanically mixed with the epoxy matrix and the resulting nanocomposite is characterized by an improvement of the compression strength (+39%) with only 0.3 wt% of GO. Furthermore, in the case of CFRPs, the approach consisting of the addition of micro/nanoparticles [23,24] to the polymer matrix, allows to remarkably improve their mechanical properties: for example, in the work of Mostovoy et al. [25], the addition of the APTES/aminoacetic acid functionalized graphene oxide induces an increase in the tensile strength and modulus of about 39% and 31%, respectively, due to the improvement of the interfacial strength between the fibers and the matrix. By adding 5 wt% of SiC NPs, Alsadi [26] increased the mode II interlaminar fracture toughness of CFRPs and aramidic fiber reinforced polymers of about 34% and 46%, respectively.

The focus of this work was to study the effects of two different carbon nanomaterials, i.e., GNPs and CNFs, on mechanical and thermal properties of a high toughness patented epoxy matrix, for space applications. These fillers were mixed by using three different dispersion techniques, respectively: a high shear rate (HSR), ultrasonication (US) and fluidized bed mixing (FBM). Moreover, in addition to the dispersion technique, another tunable parameter will be the filler typology. Therefore, the novelties of this work consist of using an industrial scale mixing technique (FBM), which allows to produce large quantities of nanocomposites with a dispersion grade comparable to the lab-scale technique (such as sonication), as well as the study of composites realized using that technique to mix fillers with different aspect ratios.

All of the realized nanocomposites were characterized in terms of thermal stability, dynamic-mechanical analysis, and fracture toughness. One nanocomposite for each filler typology was selected as a matrix to manufacture the unidirectional CFRC system. Two main criteria of selection were identified: (a) the characterization results, in particular the fracture toughness of the nanocomposite; (b) the production rate (i.e., the amount of resin achievable in a single day). The two manufactured CFRCs where mechanically tested to measure the composite interlaminar fracture toughness under mode I and mode II, to assess the effect of the filler on the delamination resistance of the composite.

## 2. Materials and Methods

### 2.1. Materials

The epoxy matrix employed in this work is the patented epoxy resin labelled HXE75 [27]. This polymer system is characterized by a viscosity (at room temperature) of about 10^4^ Pas and a minimum level of about 1 Pas at ~100 °C. This system has been developed for the purpose of obtaining the best results for the reinforcing fiber impregnation: in fact, the polymer formulation was selected, in order to guarantee a satisfactory compromise between the fluidity necessary to wet the fibers during the filament impregnation phase and the need to minimize the dripping and other manufacturing defects.

As previously stated, two different graphene based fillers have been considered in this work: graphene nanoplatelets, with commercial name “G4Nan”, produced by Nanesa (Arezzo, Italy) and graphene nanofibers, hereafter labelled as “GANF” and supplied by the Antolin Grup (Burgos, Spain). The used GNPs are characterized by a flake thickness of about 8 nm and a specific surface area of about 56 m^2^/g. According the GANFS datasheet, the nanofibers are characterized by a fiber diameter ranging from 20 to 80 nm and a specific surface area of 80–120 m^2^/g. The employed carbon fibers are the Toray T700s, characterized by a tensile strength of 4900 MPa and a tensile modulus of 230 GPa.

### 2.2. Nanocomposite Preparation

The three different mixing techniques were employed in this work, respectively: (a) the high shear rate (HSR) mixing method performed by an UltraTurrax T25 homogenizer, (b) a tip ultrasonication (US) method using a Misonix S3000 sonicator and (c) an in-house optimized fluidized bed mixing method (FBM).

In the *high shear rate mixing* (HSR), the shear forces are generated by the relative motion between the rotating and stationary elements and a complex flow field in the turbulent regime is induced. This specific motion field promotes the disintegration of the filler particles and their homogenization within the hosting fluid [28]. Since a Turrax machine was employed as a high shear rate mixer, the attained nanocomposites by this technique will be labelled with the letter “T”.

During the *ultrasonication* (US) process, the sound energy is applied to agitate the particles and homogenize the multiphasic systems. Within a fluid, a sonicator generates alternating compressions and expansion weave pressure cycles, creating a small vacuum bubble during the low pressure profile, which collapses violently during the high pressure phases as no further energy can be absorbed. This later described phenomenon, is known under the name of “cavitation” and it is mainly responsible for the mixing process of the treated system. Generally, during such processes, the ultrasonic frequencies (f > 20 kHz) are used, hence the term “ultrasonication” [29]. The nanocomposites realized by this technique will be hereafter designed with the letter “S”.

The *fluidized bed method mixing* (FBM) technique relies on the provided energy, in the form of acoustic waves, which breaks the solid clusters throughout the system. Such waves induce an oscillating thrust to the particles in the clusters, allowing to overcome the cohesive forces that bind them together and reducing the clusters dimensions. This technique is characterized by different advantages: (1) it is a dry mixing method, and consequently there are no necessarily hazardous solvents; (2) it allows mixing large amounts of powders in a single step, reducing the processing time; (3) it allows for obtaining uniform particle mixing; (4) it does not cause temperature increases of the processed system and their constituents [30]. Upon the filler mixing using the bed fluidized technique (aiming to reduce the clusters’ dimensions), the obtained mixed powders where loaded to the epoxy resin by using an industrial high shear rate mixer for the homogenization purpose. The nanocomposites realized by using that technique are labeled with the suffix “FBM” at the end of the sample name.

Figure 1 reports the fabrication procedure for the realized nanocomposites by using the three different mixing techniques T, S, and FBM, respectively. Due to the high viscosity of the hosting matrix, the degassing step (common in all the mixing procedures) is time consuming and it involves high temperatures. It is worth noting that in T and S, mixing acetone was considered as a suitable solvent to lower the system viscosity. Conversely, no solvents were employed to implement the FBM mixing as it was performed by the author within a plant production area where there were stringent requirements for eco-friendly environments and the tough regulations for a safe working environment. Performing the FBM mixing process without any solvent could be a suitable and appropriate option for the scaling up of this technique for the plant production. Upon mixing, the uncured nanocomposites were poured in a stainless steel mold, previously coated with a release agent (FREKOTE 70), and then cured, according the HXE75 cure profile (e.g., 2 h, @120 °C). The fully cured nanocomposite plates were finally cut, according to the test standards.

Table 1 reports all of the investigated nanocomposites of these research activities. To monitor the difference among the realized samples, taking into account the usage of a solvent, in the case of the HSR and US processes, a control panel, labelled HXE75 SOLV, was manufactured for comparison, implementing the HSR procedure but without the loaded fillers. The nanocomposites realized by the HSR and US have been compared with the control sample, namely HXE75_SOLV, while the reference sample for the FBM based nanocomposites is the specimen HXE75. The filler content was chosen, according to the work of Shokrieh et al. [31], which reported a maximum increase of the critical fracture toughness and the Young modulus for the nanocomposite sample loaded with 0.5 wt% of GNPs.

### 2.3. Composites Manufacturing

Carbon fiber reinforced prepregs are realized by using a tailored film-prepregging apparatus, based on a modified hot melting plant. The prepreg manufacturing process is divided into two distinct stages: (1) the filming of the resin and (2) the later impregnation of the carbon fibers. The neat or modified resin is loaded in a filming apparatus, where it is deposited on siliconized paper, through hot rollers at a suitable temperature and later stored at a low temperature. During the impregnation, the coated resin films pass through the impregnation machine embedding the unidirectional carbon fiber reinforcement (T700 UD) on both sides. The manufactured composite prepregs are stored at −4 °C before the final lamination, according the chosen layout for the specific test; the CFRP prepregs fabrication procedure is schematized in Figure 2.

### 2.4. Nanocomposites Characterization

The differential scanning calorimetry (DSC) was performed using a DSC Q1000 system (TA Instruments, New Castle, DE, USA) under a nitrogen flow (50 mL/min) and a heating ramp of 10 °C/min. The samples used had a weight of 5 ± 0.5 mg.

The thermogravimetric analyses (TGA) were conducted using a TGA Q500 system (TA Instruments, New Castle, DE, USA) and the tests were performed under a nitrogen atmosphere (50 mL/min), setting a heating ramp of 10 °C/min. The sample weight was around 10 ± 0.5 mg and the maximum temperature reached was 650 °C. The precision on the temperature measurements during each scan was ± 0.5 °C in the range of 30–400 °C and ±1 °C in the 400–800 °C range.

The dynamic mechanical analysis (DMA) was performed using Q800 DMA (TA Instruments, Milan, Italy) at a fixed frequency of 1 Hz and 3 °C/min heating ramp. Two testing configurations were set: the double cantilever mode for the precise evaluation of the glass transition and the 3-point bending mode to evaluate the storage modulus @35 °C. The nominal sample dimensions were 60 × 10 × 2.5 mm^3^ and the amplitude was set to 20 µm for all of the tests.

Mode-I fracture tests were conducted using a single edge notched beam (SENB) geometry, according to the ASTM D5045-99 standard test method. To ensure the plane strain conditions, the following specimen sample geometry was employed: 24 × 6 × 3 mm^3^. According to the standard, the crack length, *a*, was selected such that 0.45 < *a/W* < 0.55, where *W* is the sample width. The fracture tests were performed using the Lonos tenso test TT1 (Lonos Test, Monza, Italy) dynamometer equipped with 250 N load cell and at a displacement rate of 10 mm/min. The K_IC_ values were calculated, according to the standard as:(1)$$K_{\mathrm{IC}}=f\left(a/W\right)\left(\frac{P_{Q}}{{\mathrm{BW}}^{1/2}}\right)$$
with:(2)$$f\left(\frac{a}{W}\right)=6{\left(\frac{a}{W}\right)}^{1/2}\frac{\left[1.99-\left(\frac{a}{W}\right)\left(1-\frac{a}{W}\right)\left(2.15-3.93\frac{a}{W}+2.7{\left(\frac{a}{W}\right)}^{2}\right)\right]}{\left(1+2\frac{a}{W}\right){\left(1-\frac{a}{W}\right)}^{3/2}}$$
where P_Q_ is the load at failure and B is the specimen thickness. The values of G_IC_ for each nanocomposite were evaluated according the following equation:(3)$$G_{\mathrm{IC}}=\frac{U}{\mathrm{BW}\Phi }$$
where U is the corrected value of the energy and Φ is the energy calibration factor; U is obtained by subtracting U_i_ from U_Q_, i.e., the areas underneath the load-displacement curves for the cracked and un-cracked sample, respectively.

The optical micrographs of the nanocomposites were acquired by using an optical Olympus BX51 Instrument equipped with different magnitudes of oculars. The samples observed by the optical microscopy were cut to a thickness of approximately 100 µm by using a precise sawing machine operating at a high rotating speed.

### 2.5. Composites Characterization

The tensile tests were performed, according to the standard ASTMD ASTM D3039, using an MTS Landmark dynamometer (Seattle, WA, USA) equipped with a 500 KN loading cell. The glass reinforced composites were used as tabs (tabs length: 50 mm) and glued to the sample through an epoxy bi-component adhesive (Loctite EA 9497). The crosshead rate speed used was 2 mm/min and the final sample dimensions were 250 × 15 × 1 mm^3^. The flexural tests were performed, according to ASTMD D790, by way of a Lonos tenso test TT1 (Lonos Test, Monza, Italy) equipped with a 5 KN load cell. The adopted span to depth ratio was 32:1, and the crosshead speed was computed, based on the following equation:R = ZL^2^/2d(4)
where R is the crosshead speed (expressed in mm/min), Z is the rate of straining of the outer fibers (constant value 0.01), L is the support span and d is the sample thickness. The tested samples are 70 × 13 × 2 mm^3^ in size, with a 60 mm support span length.

Mode I interlaminar fracture toughness tests were carried out to measure the crack propagation energy of the CFRPs in the mode I configuration, according to the standard ASTM D5528. The opening of the Mode I interlaminar fracture toughness, G_IC_, was measured by testing the DCB (double cantilever beam) specimens and for the statistical purpose, five samples were tested from each set of materials. The aluminum loading blocks were glued on the cracked side of the sample and starting with the tip of the crack, a series of markers were drawn on the lateral side: for the first 5 mm the marker spacing was 1 mm, while the subsequent markers were 5 mm apart from each other. The loading blocks were loaded in tension, with a crosshead speed equal to 1 mm/min, and the monitoring of the crack length and specimen aperture was performed using a speed camera. The crack length is 50 mm and it was introduced into the CFPR during the lamination stage interleaving a releasing film with a 13 µm thickness. The final sample dimensions were 125 × 25 × 4 mm^3^.

The Mode II interlaminar fracture toughness of the composites was measured, according to the ASTM D7905, considering an ENF (end notched flexure) geometry. The tests were conducted with the Lonos tenso test TT1 (Lonos Test, Monza, Italy) dynamometer equipped with 5 KN load cell. The sample was loaded in a 3-point bending jig and centrally loaded with a crosshead speed of 0.5 mm/min. The pre-crack was introduced by adding a releasing film (13 µm in thickness) during the layout lamination. The sample dimensions were 130 × 25 × 4 mm^3^, with a pre-crack length of 45 mm over a bending span of 100 mm. According to the standard test [32], the compliance calibration tests were performed on the sample before the effective fracture test. Mode I and Mode II sample geometries are reported in Figure 1.

## 3. Results and Discussion

### 3.1. GNP Loaded Nanocomposites

The thermal behavior of the neat and nanocomposite samples was analyzed using DSC to assess the effect of the investigated filler over the glass transition temperature (Tg) of the hosting matrix. Table 2 reports the values of Tg for the tested samples, including the HXE75 SOLV sample, which is a neat hosting matrix, based upon the same solvent procedure by which all of the other investigated mixed nanocomposites have undergone. The reported data confirm a slight variation of Tg, compared to the solvent-free matrix, in the presence of acetone. According to Loos et al. [33], this feature can be likely attributed to a modification of the network structure, due to the solvent exposure. The addition of GNPs increases the hosting matrix Tg, regardless of the employed mixing methods (except for the HSR manufactured samples). As reported by Tang et al. [15], Tg is strongly affected by the graphene dispersion grade, and a high dispersion can cause a remarkable increase of this property. Therefore, these results can indicate that nanocomposites obtained by FDM and US reveal the optimal dispersion level, compared to other techniques, whereas the mixing operated by the high shear rate system (i.e., Turrax) shows the lowest level of disaggregation and homogeneity of the filler network. This is further supported by the work of Hielscher et al. [34] reporting a higher level of efficiency of the ultrasound mixing technique, compared to the rotor-stator high speed Turrax system for the dispersion of the solid filler in a liquid viscous polymer matrix.

Optical microscopy was employed to macroscopically evaluate the achieved dispersion level of the GNPs. The micrographs of the manufactured nanocomposites (Figure 2), in line with the results of the thermal analysis, support the conclusion that the fluidized bed mixing technique allows for obtaining a higher level of both the disaggregation and homogeneity of the graphene, inducing an exfoliation with the particle dimensions of about 10–20 µm. Conversely, the GNP clusters for the Turrax dispersed samples are much bigger, with a maximum size of about 90–100 µm.

The effect of the GNP content on the thermal stability of the epoxy system was investigated by a thermo-gravimetric analysis. Figure 3 shows the TGA results of the neat and GNP filled nanocomposites in an inert atmosphere. The thermogram of the neat resin is characterized by a single degradation step, which starts at 300 °C and terminates around 450 °C. The thermal degradation end set and the onset temperatures (T^endset^ and T^onset^, respectively) were determined by the intersection of the two tangents, and the temperature corresponding to the peak of the weight loss derivative curve identifies the maximum degradation temperature (T^MAX^). As shown in Figure 3, the addition of GNPs does not affect remarkably the thermal stability of the hosting matrix for the nanocomposites mixed by Turrax and by ultrasonication, as well as their char yield (i.e., the weight residual at 600 °C) Conversely, for the samples realized using the FBM mixer, it is evident that the addition of GNPs slightly increases the char yield, as expected, in agreement with the result published by Liu et coworkers in [35]. As a matter of fact, the presence of the well dispersed GNPs should increase the thermal conductivity of the hosting matrix, limiting the degradative processes and also inducing an increase of the weight residual. Moreover, analogously to this work, the presence of this filler (at content ≤ 0.5 wt%) slightly increases the thermal stability of the system, with a small rise of T^MAX^, compared to the neat matrix. As the decomposition temperature could be directly related to the crosslink density of the thermoset polymers, the T^MAX^ increase is likely attributed to the non-stoichiometry following the GNPs incorporation in the epoxy matrix [36].

The DMA tests were performed to evaluate the storage modulus of the nanocomposites, as a function of the temperature. The main results, in terms of the modulus @35 °C, and Tg for the neat epoxy and the GNP loaded nanocomposites are reported in Table 3 and Figure 4. For the systems processed by the HSR and US, the filler presence unremarkably influences the dynamical mechanical behavior of the original matrix, with a negligible change of about 3% of the modulus. The modest variation is measured in the case of the FBM mixed nanocomposite (~7%), however the DMA Tg undergoes a more relevant increase of about 4.5 °C. This behavior can be attributed, once again, to the better dispersion of GNP in the system mixed with FBM, compared to the other systems, confirming and also supporting the results of Le et al. [37], who reports a slight enhancement of the thermomechanical properties of the GNP loaded polymer, compared to other graphene based fillers. Towards this result, it is worth noting that only in the case of a strong interaction between the filler and the matrix, a remarkable increase of the storage modulus is expected, due to the enhancement of the stress transfer mechanism at the interface, as in the case of the graphene oxide based nanocomposites [38].

The mode I fracture toughness results measured by the SENB tests are summarized in Table 3 and shown in Figure 5. All samples showed a significant improvement in the fracture toughness properties, compared to the pristine epoxy samples. The largest variation of G_IC_ was observed for the sample achieved by ultrasonication (HXE75 + 0.5 wt%_G_S), with an increase of 32.9%, which leads to a final value of 1.01 KJ/m^2^. This result can be explained considering the GNP dispersion in terms of both the particle desegregation and the uniformity within the hosting matrix. In fact, the samples HXE75 + 0.5 wt%_G_T are characterized by a worse overall dispersion, compared to the HXE75 + 0.5 wt%_G_S specimens (as observed in Figure 2), thus limiting the reinforcement efficiency of the GNPs in the hosting matrix. This conclusion is well in accordance with the work of Domun et al. [39], who observed that an increase in the GNP content will induce a worsening of the dispersion degree (due to agglomeration) with the consequent detrimental effect over the fracture toughness performance. Concerning the sample HXE75 + 0.5 wt%_G_FBM, although this set reveals a higher level of dispersion, compared to the sonicated one, the larger particles/clusters act more significantly as crack deflectors and crack bridging elements improving the toughness, as also suggested by Chatterjee et al. [40], who demonstrated that larger GNPs induce a higher increase of the fracture toughness in the epoxy matrix, also due to their larger aspect ratio.

### 3.2. GANF Loaded Nanocomposites

Table 4 reports the thermal characterization of the GANF based nanocomposites and the corresponding optical images are shown in Figure 6. The samples loaded with GANFs mixed using HSR system show a negligible variation of the glass transition temperature, analogously to the results obtained for the HXE75 + 0.5 wt% G_T sample, due to the poor dispersion achieved by the technique, as reported in Figure 6a. On the contrary, the nanocomposites obtained by the FBM and US procedures show an increase of Tg of about 3.1 °C and 2.6 °C, respectively. These results can be related by the considerable reduction of the final filler cluster formation, as supported by the optical micrographs in Figure 6b,c. The higher level of dispersion achieved by the different investigated mixing techniques can reflect the nanocomposites’ thermal stability results, hereafter reported and discussed. Referring to Table 4 and the thermograms reported in Figure 7, the presence of GANFs does not affect the thermal stability of the hosting matrix, in terms of T^Onset^, T^offSet^, and T^MAX^, with only a slight increase of the char yield for the FBM mixed nanocomposites. This is in accordance with the results published by Choi et al. in [41], where the limited carbon nanofiber content does not significantly affect the thermal stability of the hosting matrix, as in our cases for the sonicated and fluidized mixed systems. Whereas, in the case of the HXE75 + 0.5 wt% GANF_T nanocomposite, the poor level of the achieved dispersion can detrimentally affect the degradation of the original resin.

The DMA results, reported in Table 5 and Figure 8, still support the conclusion achieved by discussing the thermal properties: in fact, the poor dispersion attained in the case of HXE75 + 0.5 wt% GANF_T prompt the detrimental effect over the mechanical performance, compared to the original resin and the solvent-treated neat system. As a matter of fact, no significant variation of Tg is reported for this nanocomposite set, whereas the HSR and US based mixed nanocomposites reveal a Tg increase of about 6.7 °C and 3.2 °C, respectively. The GANF based nanocomposites were also characterized, in terms of the fracture toughness through the SENB test. The HSR processed nanocomposites show a lower K_IC_ and G_IC_, compared to the neat matrix, and that behavior is still attributed to the lower degree of dispersion achieved, if compared to the other mixing methods. It clearly demonstrates that GANFs are not suitable filler to efficiently enhance a hosting matrix by using a high rate shear technique, as this procedure is not sufficient to provide the necessary energy level to break the nanofiber clustering agglomeration and the uniform distribution in space. This conclusion is inherently linked with the large surface area of the GANFs (>1400 m^2^/g [42]), which induces remarkable van der Waals forces among the nanofibers, hindering the cluster disintegration. Conversely, the GNPs are characterized by a lower surface area (~300 m^2^/g [43]), and consequently they are energetically weaker and more inclined to exfoliate and to disperse within the matrix, compared to the GANF fillers. The homogeneous dispersion of GANF within the hosting epoxy using the US and FBM techniques justifies the remarkable increase of both K_IC_ and G_IC_, which is also in agreement with other published works, such as Bortz et al. [44]. A significant improvement of the fracture toughness performance is achieved in the case of the HXE75 + 0.5 wt% GANF_FBM nanocomposite, with an increase in the critical stress intensity and the critical energy release rate of about 9.5% and 29.2%, respectively, when compared to the neat matrix, as reported in Table 5 and Figure 9.

### 3.3. Composites Chatacterization

Among the six differently loaded nanocomposite matrices which were prepared and investigated, only two of them were chosen to manufacture the final long carbon fiber composites and to characterize them for a later comparison with a neat matrix-based carbon reinforced composite configuration. Two main factors were analyzed as the passing criteria for the composite specimens, respectively, the fracture toughness results and the large-scale production capability. This later feature is associated with the potentiality of making available a large amount of the filled matrix for the manufacturing of extended components or structures, such as for the aeronautical or aerospace industry. As previously stated, FBM allows for producing large amounts of a modified epoxy matrix in a reasonable amount of time, in an automatic, unattended man-working process and at the same time investing reduced resources, in terms of energy. Conversely to the high shear rate and the ultrasonication methods, as already mentioned, this technique was employed without solvent or viscosity modifier, and thus it stands as a very interesting industrial process setting, which assure a safe working environment and more environmental protection. Contrariwise, HSR and US allow for working with laboratory scale amounts of materials (maximum 500 g of resin), and a feasibility study for a potential scaling-up methodology is indeed complex and costly for a large production. Specifically regarding this research, due to the high viscosity of the HXE75 matrix, both the HSR and US techniques require the use of a solvents, such as acetone, and thus increase the potential hazard and degree of pollution for the implemented process. Table 6 outlines the criteria for the selection of the most suitable efficient technique to manufacture the loaded matrix for the later investigation of carbon fiber reinforced systems.

Based on the data reported in Table 6, the results clearly show that the most suitable nanocomposite systems to consider for UD composites are the HXE75 + 0.5 wt% G_FBM and HXE75 + 0.5 wt% GANF_FBM. In the following, the composite sets were identified with labels, such as HXE75 + 0.5 wt% G_FBM_UD and HXE75 + 0.5 wt% GANF_FBM_UD, where the suffix UD indicates the unidirectional orientation of the T700 Toray carbon reinforcement within the single prepreg layer.

The tensile tests were performed, in order to analyze the effect of the chosen fillers (i.e., GNP and GANF) on the composite mechanical properties. Figure 10 reports the test results in terms of the tensile strength and modulus. The addition of the fillers induced, in both cases, a reduction of the tensile modulus for both the GNP and GANF loaded composites, respectively, of about −7.4% and −5.2% and a tensile strength improvement of +6.1% and +8.4%. The analogous effect was reported by Palmeri et al. [45], studying the effect of CNF on the mechanical properties of a carbon fiber reinforced composite, toughened with a triblock elastomer; the authors attributed the degradation of the tensile modulus to the agglomeration of the CNFs that reduces the stress transfer between the matrix and the carbon fibers. The increase in the tensile strength is likely associated with the increased crack propagation resistance by the bridging effect, due to the GNP and GANF presence, which improves the composite strength [46].

Flexural rigidity is one of the main features describing the mechanical behavior of laminated composite materials. Now highlighted in Figure 11, the addition of both filler typologies lead to a positive change of the flexural strength (about 6.5% and 4.3%, respectively, for the GNP and GANF loaded composites) and no appreciable changes in the flexural modulus. The increase in strength can be attributed to the improvement of the mechanical properties of the interface between the fiber and matrix, allowing for storing more energy before breaking and promoting the fiber breakage, rather than the fiber-matrix interface fracture. That behavior is in agreement with the work of Tareq et al. [47], who studied the effect, individually or combined, of the GNPs and nanoclays on the mechanical and thermal properties of the carbon reinforced epoxy composites: the authors showed that GNPs induce a more pronounced increase of the flexural strength, while the improved of stiffness nanoclays are more effective.

Figure 12 reports the mode I and II fracture toughness test results for the neat and the nanomatrix reinforced carbon composite laminates. G_IC_ is positively affected by the GNPs’ and GANFs’ presence, with an increase of about 33.1% and 14.3%, respectively. The Mode I fracture toughness reflects the interaction between the epoxy matrix and the fiber reinforcement, giving a safer result for the total behavior of the composite in the fracture [48]. The improvement of the G_IC_ values due to the filler content confirms the conclusion that the fiber/matrix interface is enhanced, as also previously observed for the tensile and the flexural strengths. The mechanisms responsible for the increase of the polymer fracture toughness, due to the incorporation of micro/nano fillers, have been extensively studied in the last decades [49]. Brittle polymers [50], such as thermosetting resins, benefit more from the presence of micro and nanofillers, compared to the thermoplastic polymers. The main micromechanical mechanisms leading to an improvement of the fracture toughness are [51]: (1) the localized matrix plastic deformation and the void nucleation, (2) filler debonding, (3) crack deflection, (4) crack pinning, (5) fiber pull-out, (6) crack tip blunting and (7) particle/fiber deformation or brakeage. In the composite laminates, the crack propagation suppression is mainly due to the crack deflection: when the crack begins to grow, it deflects, due to the GNPs’ and GANFs’ presence, consequently suppressing the crack growth [42]. For what concern the Mode II fracture toughness, it is clear, according to Figure 12, that the addition of fillers induces a detrimental effect with a reduction of G_IIC_, compared to the neat matrix, of about −47.6% and −35.8% for the GNP and GANF loaded composites, respectively. That behavior was already observed by Ahmadi-Moghadam et al. [52], who studied carbon fibers reinforced composites loaded with GNPs functionalized in different ways. The referred work reports the experimental evidence of the G_IIC_ reduction in the un-modified GNP loaded epoxy. This statement is supported by the literature [53], according to which the nanocomposites based on the epoxy matrix loaded with un-modified GNPs are characterized by a lower Mode II fracture toughness, compared to the pristine matrix. The results of composites’ mechanical characterizations are summarized in Table 7.

## 4. Conclusions

The effect of the mixing technique on the thermal and mechanical properties of a GNP and GANF loaded epoxy resin were investigated and reported. Three different dispersions and mixing methodologies were employed, respectively, the high shear rate (HSR), ultrasonication (US), and the fluidized bed method (FBM). The HSR and US mixing were performed, considering a dispersant solvent, such as acetone, whereas the FBM was characterized by a solvent-free approach fulfilling the requirement of a potentially scalable technique and safer processing conditions. The TGA analysis revealed a negligible effect on the thermal stability, due to the fillers presence, regardless of the mixing technique. Optical microscopy highlighted that the optimal dispersion, in terms of the homogeneity and cluster size, is achieved for the samples mixed using US and FBM and that observation is supported by the DSC and DMA results, which show that these nanocomposites are characterized by the largest Tg increase. The addition of both filler typologies induces a global increase of the fracture toughness, in term of K_IC_ and G_IC_, compared to the neat matrix: the largest increase of G_IC_ is obtained for the samples mixed with US (+32.9% and 10.5% for the GNP and GANF nanocomposites) and the FBM (+27.8% and 29.2% for the GNP and GANF nanocomposites).

Analyzing the fracture toughness results and taking into account the large-scale production capability of the mixing technique, which is a fundamental requirement for the industrial scaled production, the FBM technique was chosen to manufacture the carbon reinforced composites (CFRCs). Two CFRC laminates were manufactured, one for each filler, and were characterized in terms of tensile, flexural, and interlaminar fracture toughness. The tensile tests have shown an increase in the tensile strength of 6.1% and 8.5% for the GNP and GANF loaded CFRCs, respectively. Furthermore, the flexural strength undergoes an improvement due to the GNP and GANF reinforcement, with an increase of about 6.5% and 4.3%, respectively. For what concerns the interlaminar fracture toughness, Mode I is improved by the filler presence, while Mode II undergoes a drastic reduction, likely due to the reduction of Mode II the fracture toughness of the epoxy matrix after the addition of GNPs and GANFs, as reported in the literature.

## Data Availability

Not applicable.

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
