# Peer review of "Effect of the Mixing Technique of Graphene Nanoplatelets and Graphene Nanofibers on Fracture Toughness of Epoxy Based Nanocomposites and Composites"

_polymers, 2022, doi:10.3390/polym14235105_

Round 1

Reviewer 1 Report

The effect of the method of obtaining dispersion of various nanofillers in epoxy matrix described by the authors of the work on their properties is quite interesting and corresponds to the current trends in obtaining nanocomposites. However, the article requires correction and clarification of the obtained results. Below is attached a list of major and minor remarks that need to be fulfilled and revised in order for the work to be accepted for publication.

The authors should supplement the introduction with relevant literature sources, 15 of the 21 references in the introduction are older than 5 years. In addition, provide more disclosure on the effects of different nanoparticles. The following references can be included in the "Introduction" part to improve the quality of the manuscript, as they provide the necessary information:

https://doi.org/10.3390/polym14020338

https://doi.org/10.1002/pc.26193

https://doi.org/10.1007/s11029-022-10004-7

https://doi.org/10.1134/S1070427221080097

https://doi.org/10.1016/j.compositesb.2021.109218

https://doi.org/10.3390/polym13234116

Some details were missed in the description of the study objects. Add the properties of the epoxy matrix according to the data sheet and specify the hardener used to cure the epoxy system.

In line 166 the sentence: "Obtained micrographs, for these samples, have revealed a good dispersion of the fillers despite the high viscosity of the hosting matrix likely due to the efficiency of the performed FBM premixing stage." describes the obtained results, which is not required in section 2.2.

Scheme 1 shows the process of making an unfilled composite? The process of making a fibre-reinforced composite should be added on scheme.

Specify dimensions of the specimens tested on tension and bending. Line 230-235

If possible, describe the technique of clamping the specimens for tensile and bend testing (this comment is in the personal interest of the reviewer and may not be reflected in the article).

In line 269 referring to [28] you point out a possible change in the glass transition temperature due to changes in the three-dimensional cross-linked structure of the composite. However, in this work the curing is carried out at room temperature and acetone remains in the composite structure, whereas in your work the curing is carried out at temperatures up to 120 â—‹C resulting in complete removal of acetone, which is confirmed by the non-worsened TGA data.

In line 302 to reveal in more detail by text the reason for the increase in Char Yield of the composition HXE75+0.5wt%_G_FBM.

In line 527 In your article an increase in mode II fracture toughness for unreinforced matrices is noted, is it possible to explain the decrease in this parameter on reinforced composite by referring to the decrease in this matrix parameter in other works.

Author Response

REVIEWER 1

The effect of the method of obtaining dispersion of various nanofillers in epoxy matrix described by the authors of the work on their properties is quite interesting and corresponds to the current trends in obtaining nanocomposites. However, the article requires correction and clarification of the obtained results. Below is attached a list of major and minor remarks that need to be fulfilled and revised in order for the work to be accepted for publication.

  • The authors should supplement the introduction with relevant literature sources, 15 of the 21 references in the introduction are older than 5 years. In addition, provide more disclosure on the effects of different nanoparticles. The following references can be included in the "Introduction" part to improve the quality of the manuscript, as they provide the necessary information:

https://doi.org/10.3390/polym14020338

https://doi.org/10.1002/pc.26193

https://doi.org/10.1007/s11029-022-10004-7

https://doi.org/10.1134/S1070427221080097

https://doi.org/10.1016/j.compositesb.2021.109218

https://doi.org/10.3390/polym13234116

We thank the reviewer for valuable suggestions: some of the indicated papers were added to the manuscript and commented.

  • Some details were missed in the description of the study objects. Add the properties of the epoxy matrix according to the data sheet and specify the hardener used to cure the epoxy system.

We thank the reviewer for the observation. The considered matrix is patented by the supplier and it is non-commercial epoxy resin, and all the relevant properties were added in the HXE75 description within the materials and methods section.  Moreover, the resin is a single compound system and thus no hardener is supposed to be added to achieve final polymerized state.

  • In line 166 the sentence: "Obtained micrographs, for these samples, have revealed a good dispersion of the fillers despite the high viscosity of the hosting matrix likely due to the efficiency of the performed FBM premixing stage." describes the obtained results, which is not required in section 2.2.

We thank the reviewer for the valuable suggestion: we have removed the aforementioned sentence.

  • Scheme 1 shows the process of making an unfilled composite? The process of making a fibre-reinforced composite should be added on scheme.

We thank the reviewer for the suggestion. The description of the unfilled composited is reported in the text and don’t need a specific scheme. We have added the scheme 2 that describe the CFRP production.

  • Specify dimensions of the specimens tested on tension and bending. Line 230-235

We thank the reviewer for highlighting this missing data. We have added sample dimensions.

  • If possible, describe the technique of clamping the specimens for tensile and bend testing (this comment is in the personal interest of the reviewer and may not be reflected in the article).

In tensile tests, tabs are glued to the CFRP and the samples are clamped using hydraulic clamps equipping the dynamometer machine at specific pressure. Before clamping, at middle position of the sample gage length, a strain gauge is applied with an alignment in the fiber direction to measure precisely the strain value and then to compute the tensile modulus. Flexural tests are performed using a 3-point binding configuration, without further preparation.

  • In line 269 referring to [28] you point out a possible change in the glass transition temperature due to changes in the three-dimensional cross-linked structure of the composite. However, in this work the curing is carried out at room temperature and acetone remains in the composite structure, whereas in your work the curing is carried out at temperatures up to 120 â—‹C resulting in complete removal of acetone, which is confirmed by the non-worsened TGA data.

We thank the reviewer for the interesting question. According to our statement, the Tg variation is not attributed to the acetone presence after the crosslinking process, which is clearly evaporated completely upon 2h at 120°C, but likely related to the effect induced by the acetone content on the epoxy matrix during network forming process until the gelation stage. Although the boiling point of acetone is ~60°C, also at moderate vacuum pressure, it is reasonable to assume that a small quantity of acetone gets trapped (due to hydrogen bonds) among the epoxy chains also after gelation (which occurs after a few minutes at the cure temperature). Having said that, after gelation, the residual acetone leaves likely induce a small increase of the free volume with consequent variations of thermal and themo-mechanical properties.

  • In line 302 to reveal in more detail by text the reason for the increase in Char Yield of the composition HXE75+0.5wt%_G_FBM.

We thank the reviewer for the question. As reported within the manuscript, the presence of well dispersed GNPs increases the thermal conductivity of the matrix, delaying the thermal degradation of the system (with consequent small increase of TMAX); moreover, GNPs presence allows obtaining a more compact carbonaceous char, which hinders the macromolecular chains degradation and the removal of degradation compounds, with consequent higher weight residual at 700°C

  • In line 527 In your article an increase in mode II fracture toughness for unreinforced matrices is noted, is it possible to explain the decrease in this parameter on reinforced composite by referring to the decrease in this matrix parameter in other works.

We thank the reviewer for the question. As reported in the text, the decrease of this parameter is attributable to the usage of an un-modified carbon nanofiller rather than a functionalized one, as reported in other works. The absence of a chemical link between filler and matrix limits the efficiency in toughening performances of the fillers that, in mode II tests, can act as a defect rather than a reinforcement.

Reviewer 2 Report

The entitled article “Effect of mixing technique of graphene nanoplatelets and graphene nanofibers on fracture toughness of epoxy based nanocomposites and composites” was carefully reviewed.

Aldobenedetto Zotti and co-authors demonstrate interesting work.

It needs revision before consideration for publication in the Polymers.

 Limitation of the work, novelty and contributions should be highlighted more.

 The introduction still needs to be improved. The following references are relevant to the nanocomposite & their various applications, which should be accommodated in the introduction section to improve the quality of manuscript.

It is better to rewrite properly by referencing the references below.

Materials Science for Energy Technologies 4, 92-99, 2021. Materials Science for Energy Technologies 4, 107-112, 2021. Polymers 202214(21), 4516.

Molecular weight of the Epoxy polymer need to be mentioned.

The provided Optical microscopy images need full scale with high magnification details.

It is better to compare the cost of the total reaction condition to be compared.

The conclusion section seems awkward and should be rephrased in a more catchy way.

Still in the current state, there are some typographical errors. Therefore, the authors are advised to recheck the whole manuscript.

Overall the article is fairly well written, after addressing the above comments the article may be considered for publication.

Author Response

REVIEWER 2

The entitled article “Effect of mixing technique of graphene nanoplatelets and graphene nanofibers on fracture toughness of epoxy based nanocomposites and composites” was carefully reviewed.

Aldobenedetto Zotti and co-authors demonstrate interesting work.

It needs revision before consideration for publication in the Polymers.

  • Limitation of the work, novelty and contributions should be highlighted more.

We thank the reviewer for the observation: we have added the following period to clarify the goal of the work:

“therefore, the novelties of this work consist of using an industrial scale mixing technique (FBM), which allows to produce large quantity of nanocomposites with a dispersion grade comparable to lab-scale technique (such as sonication), as well as the study of composites realized using that technique to mix fillers with different aspect ratios.”

  • The introduction still needs to be improved. The following references are relevant to the nanocomposite & their various applications, which should be accommodated in the introduction section to improve the quality of manuscript.

It is better to rewrite properly by referencing the references below.

Materials Science for Energy Technologies 4, 92-99, 2021.

Materials Science for Energy Technologies 4, 107-112, 2021.

Polymers 2022, 14(21), 4516.

We thank the reviewer for valuable suggestions: some of the suggested papers were added to the manuscript.

  • Molecular weight of the Epoxy polymer need to be mentioned.

As reported in the text, the matrix is a patented epoxy resin used in aeronautical applications. For this reason, we have no way to obtain its Epoxy Equivalent Weight.

  • The provided Optical microscopy images need full scale with high magnification details.

We thank the reviewer for the observation. We have add full scale and optical zooms.

  • It is better to compare the cost of the total reaction condition to be compared.

We are not able to provide that analysis.

  • The conclusion section seems awkward and should be rephrased in a more catchy way.

We thanks the author and we have modified the conclusion section.

  • Still in the current state, there are some typographical errors. Therefore, the authors are advised to recheck the whole manuscript.

We thank the author for the attention: we have performed the indicated corrections.

Overall the article is fairly well written, after addressing the above comments the article may be considered for publication.

Round 2

Reviewer 1 Report

The authors considered most of the comments or adequately responded to the remarks contained in the review; therefore, the work may be approved for publication after editorial and English corrections.

Reviewer 2 Report

The revised article  “Effect of mixing technique of graphene nanoplatelets and graphene nanofibers on fracture toughness of epoxy based nanocomposites and composites” was well revised, it may be acceptable for publication in the present form.